# Reduced form of Galectin-1 Suppresses Osteoclastic Differentiation of Human Peripheral Blood Mononuclear Cells and Murine RAW264 Cells In Vitro

**DOI:** 10.3390/biom14010121

**Published:** 2024-01-17

**Authors:** Tomoharu Takeuchi, Midori Oyama, Mayumi Tamura, Yoichiro Arata, Tomomi Hatanaka

**Affiliations:** 1Faculty of Pharmacy and Pharmaceutical Sciences, Josai University, 1-1 Keyakidai, Sakado 350-0295, Saitama, Japan; oyamami@josai.ac.jp (M.O.); tmmhtnk@josai.ac.jp (T.H.); 2Faculty of Pharma-Science, Teikyo University, 2-11-1 Kaga, Itabashi, Tokyo 173-8605, Japan; m-tamura@pharm.teikyo-u.ac.jp (M.T.); arata@pharm.teikyo-u.ac.jp (Y.A.); 3School of Medicine, Tokai University, 143 Shimokasuya, Isehara 259-1193, Kanagawa, Japan

**Keywords:** osteoclast, Gal-1, oxidation, differentiation, bone, osteoporosis

## Abstract

Galectin-1 (Gal-1) is an evolutionarily conserved sugar-binding protein found in intra- and extracellular spaces. Extracellularly, it binds to glycoconjugates with β-galactoside(s) and functions in various biological phenomena, including immunity, cancer, and differentiation. Under extracellular oxidative conditions, Gal-1 undergoes oxidative inactivation, losing its sugar-binding ability, although it exhibits sugar-independent functions. An age-related decrease in serum Gal-1 levels correlates with decreasing bone mass, and Gal-1 knockout promotes osteoclastic bone resorption and suppresses bone formation. However, the effect of extracellular Gal-1 on osteoclast differentiation remains unclear. Herein, we investigated the effects of extracellular Gal-1 on osteoclastogenesis in human peripheral blood mononuclear cells (PBMCs) and mouse macrophage RAW264 cells. Recombinant Gal-1 suppressed the macrophage colony-stimulating factor and receptor activator of nuclear factor-κB ligand-dependent osteoclast formation, actin ring formation, and bone-resorption activity of human PBMCs. Similar results were obtained for RAW264 cells. Gal-1 knockdown increased osteoclast-like cell formation, suggesting that it affected differentiation in an autocrine-like manner. Oxidized Gal-1 slightly affected differentiation, and in the presence of lactose, the differentiation inhibitory effect of galectin-1 was not observed. These findings suggest that extracellular Gal-1 inhibits osteoclast differentiation in a β-galactoside-dependent manner, and an age-related decrease in serum Gal-1 levels may contribute to reduced osteoclast activity and decreasing bone mass.

## 1. Introduction

The regulation of bone metabolism is orchestrated by the balance between osteoclastic bone resorption and osteoblastic bone formation [1]. The disruption of this equilibrium, for instance, with the overdominance of bone resorption by osteoclasts, can lead to pathological conditions such as osteoporosis, which is of particular concern in an aging society. Osteoclasts are the only cells derived from the monocyte/macrophage lineage capable of resorbing bone, and their differentiation is regulated by the macrophage colony-stimulating factor (M-CSF) and receptor activator of nuclear factor-κB ligand (RANKL) [2]. In the differentiation process, M-CSF induces the expression of the receptor activator of nuclear factor-κB, the receptor for RANKL. Subsequently, RANKL activates its downstream signaling pathways, leading to the activation of the nuclear factor of activated T-cell cytoplasmic 1 (NFATc1). NFATc1 serves as the master transcriptional regulator of osteoclast differentiation, prompting the expression of osteoclast-specific genes, including tartrate-resistant acid phosphatase (TRAP), cathepsin K, matrix metallopeptidase 9 (MMP-9), and NFATc1. The stimulated mononuclear cells then undergo cell fusion to form TRAP-positive multinuclear osteoclasts, which bind to the bone surface through polymerized actin fibers called actin rings and resorb the bone [3].

Galectin-1 (Gal-1) is an evolutionarily conserved β-galactoside-binding protein found intracellularly and extracellularly. It is synthesized as a cytosolic protein and can function in intracellular spaces [4,5]. However, it is secreted by nonclassical pathways, such as necroptosis [6] or presently undetermined pathways, and functions extracellularly in various biological phenomena, such as immunity, cancer, and differentiation [4,7]. Under extracellular oxidative conditions, Gal-1 may form intra- or intermolecular disulfide bonds and undergo oxidative inactivation, in which the oxidation of Cys^2^ is thought to be a key process, resulting in the loss of its sugar-binding ability [8,9,10]. Such oxidative inactivation may be part of the redox regulation of Gal-1 [11]. However, the oxidized form of Gal-1 also plays a role in axonal regeneration [12] and macrophage activation [13] in a sugar-independent manner.

Gal-1 plays a role in bone homeostasis. Gal-1 added extracellularly induces the osteoblastic proliferation and differentiation of human bone marrow stromal cells [14]. Gal-1 knockout in mice causes bone loss in vivo, and recombinant Gal-1 proteins suppress the osteogenic differentiation of bone marrow stromal cells in vitro [15]. In contrast, Gal-1 mRNA expression in mature osteoclasts is lower than that in progenitor monocytes, and osteoclasts derived from Gal-1-knockout mice have higher bone resorption activity than those from the wild type in vitro [16]. Age-related decreases in serum Gal-1 levels correlate with decreases in bone mass in mice and humans [17].

The result using Gal-1 knockout mice revealed that endogenous Gal-1 suppresses osteoclast activity [16]. However, given that Gal-1 functions intracellularly and extracellularly, its oxidized form also exhibits an intrinsic function; the effect of extracellular Gal-1 on osteoclast differentiation, including bone resorption activity, remains unclear. Therefore, in the present study, we investigated the effect of extracellular Gal-1 on osteoclastic differentiation, including the bone resorption activity of human peripheral blood mononuclear cells (PBMCs) and mouse macrophage RAW264 cells, widely used model cells for osteoclast differentiation [18,19], and reported that extracellular Gal-1 suppressed the differentiation of osteoclast in a β-galactoside-dependent manner.

## 2. Materials and Methods

### 2.1. Cell Culture

Noncharacterized normal human PBMCs were obtained from Fujifilm Wako (Osaka, Japan) and cultured in minimum essential medium alpha (MEMα) (Fujifilm Wako) supplemented with 10% heat-inactivated fetal bovine serum (Thermo Fisher Scientific, Waltham, MA, USA) and 1× penicillin/streptomycin (Fujifilm Wako). The mouse macrophage-like RAW264 cell line [20] was obtained from the RIKEN Cell Bank (Tsukuba, Japan) and maintained in MEMα medium containing 10% heat-inactivated fetal bovine serum and 1× penicillin/streptomycin under a humidified atmosphere containing 5% CO_2_ at 37 °C.

### 2.2. Preparation of Recombinant Gal-1 Proteins

Human Gal-1C2S, a Cys^2^-to-Ser-substituted form of the wild type, which is thought to be resistant to the oxidation-dependent inactivation of its carbohydrate affinity without affecting carbohydrate-binding ability [8] and mouse Gal-1C2S were expressed in *Escherichia coli* and affinity-purified using an asialofetuin-immobilized Sepharose column, as described previously [21]. For the mouse Gal-1 (mGal-1) wild-type-expressing plasmid, an artificial gene encoding the mGal-1 protein with optimized codon usage was synthesized by Fasmac (Kanagawa, Japan) and subcloned into the *Nde*I and *Bam*HI sites of the pET21a vector. The resultant pET-mGal-1WT plasmid was used for protein expression as described above. Contaminated lipopolysaccharide was removed from the affinity-purified protein dissolved in phosphate-buffered saline (PBS; 8.1 mM Na_2_HPO_4_, 1.47 mM KH_2_PO_4_, 137 mM NaCl, and 2.68 mM KCl; pH 7.4) using Detoxi-Gel Endotoxin Removing Columns (Thermo Fisher Scientific, Waltham, MA, USA), and the removal was checked using Limulus ES-2 Single Test Wako 0.03 (Fujifilm Wako), as described previously [22]. Heat-inactivation of recombinant Gal-1 protein was performed by boiling it for 5 min.

### 2.3. Osteoclastic Differentiation

Human PBMCs were seeded (1 × 10^5^ live cells/well) on a 96-well plate, 96-well black plate (Thermo Fisher Scientific), or Osteo Assay Surface Stripwell Microplate (Corning, NY, USA). The cells were then treated with 100 ng/mL glutathione S-transferase (GST)-tagged soluble RANKL (sRANKL) (Oriental Yeast, Tokyo, Japan) and 25 ng/mL human M-CSF (PeproTech, Rocky Hill, NJ, USA) in the presence of 10 µg/mL recombinant human Gal-1C2S protein and allowed to differentiate for 7 days for the cell viability assay and TRAP staining or for 14 days for the actin staining and resorption assay. The medium was replenished every three or four days.

Mouse macrophage RAW264 cells were seeded (1 × 10^3^ live cells/well) on a 96-well plate, 96-well black plate (Thermo Fisher Scientific), or Osteo Assay Surface Stripwell Microplate (Corning) and cultured for 1 day. The cells were then treated with 250 ng/mL GST-tagged sRANKL, as described previously [23], in the presence of 10 µg/mL recombinant mGal-1 protein and 10 mM lactose or maltose and allowed to differentiate for 4 or 14 days for the resorption assay. The medium was replenished every three or four days.

### 2.4. RNA Interference (RNAi)

RAW264 cells were seeded in a 96-well cell culture plate at a density of 1000 cells/well. After 24 h, small interfering RNA (siRNA) was transfected using Lipofectamine RNAiMAX (Thermo Fisher Scientific) according to the manufacturer’s instructions. The following siRNAs were used: *Gal-1* siRNA #1, 5′-GGG CUU AAG UGA CCA CAG Att-3′ and 5′-UCU GUG GUC ACU UAA GCC Ctc-3′; *Gal-1* siRNA #2, 5′-GAC CGC AUU UGU CUU AAU Att-3′ and 5′-UAU UAA GAC AAA UGC GGU Ccg-3′; and negative control siRNA (Silencer select negative control siRNA #1; Thermo Fisher Scientific). After 1 day, the medium was replaced, and the cells were stimulated with 250 ng/mL sRANKL (Oriental Yeast, Tokyo, Japan). The cells were further cultured for 4 days to allow differentiation and then subjected to a cell viability assay or TRAP staining. For real-time polymerase chain reaction (PCR), RAW264 cells were seeded at a density of 4000 cells/well and incubated for 24 h, followed by transfection with siRNA. Following incubation for 24 h, the cells were subjected to real-time PCR analysis.

### 2.5. Cell Viability Assay

Cell viability was assessed using Cell Counting Kit-8 (Dojindo, Kumamoto, Japan) for PBMCs or using the PrestoBlue™ cell viability reagent (Thermo Fisher Scientific) for RAW264 cells according to the manufacturer’s instructions.

### 2.6. TRAP Staining

Differentiated cells were washed with PBS and treated with 4% paraformaldehyde solution (Fujifilm Wako) for 10 min at room temperature. The cells were washed again with PBS and then stained with a TRAP staining solution containing 50 mM sodium tartrate, 45 mM sodium acetate, pH 5.2, 0.1 mg/mL naphthol AS-MX phosphate (Sigma-Aldrich, St. Louis, MO, USA), and 0.6 mg/mL fast red violet LB (Sigma-Aldrich) at pH 5.2 for 1 h or longer at room temperature. The stained cells were observed under the EVOS XL Core microscope (Thermo Fisher Scientific), and TRAP-positive cells that stained red and contained three or more nuclei were counted.

### 2.7. Actin Staining

Differentiated cells on a 96-well black plate were washed with PBS and treated with 4% paraformaldehyde solution for 10 min at room temperature. The cells were washed with PBS and treated with PBS containing 0.1% TritonX-100 for 5 min at room temperature. After washing with PBS, the cells were stained using ActinGreen™ 488 ReadyProbes™ Reagent (Thermo Fisher Scientific) dissolved in PBS for 30 min at room temperature according to the manufacturer’s instructions. The cells were washed again with PBS and stained with 1 µg/mL of -Cellstain^®^- DAPI solution (Dojindo, Kumamoto, Japan) dissolved in PBS for 5 min at room temperature. Green and blue fluorescence emitted by the stained cells was observed and visualized using a BZ-X800 fluorescence microscope (Keyence, Osaka, Japan). Images were captured using a 4× objective lens. The area surrounded by actin rings with three or more nuclei in each image was manually selected and quantified using ImageJ software (ver. 1.54d) [24].

### 2.8. Real-Time Reverse Transcription PCR

RNA extraction from differentiated cells and cDNA synthesis were performed using Power SYBR^®^ Green Cells-to-CT™ Kit (Thermo Fisher Scientific) according to the manufacturer’s instructions. Briefly, the differentiated cells were washed with PBS and lysed using a lysis solution containing DNase I. A portion of the lysate was used for reverse transcription with random primers and oligo-dT. Then, synthesized cDNA was used for real-time PCR with StepOnePlus Real-Time PCR System (Thermo Fisher Scientific) and PowerUp SYBR^®^ Green PCR Master Mix (Thermo Fisher Scientific).

All PCR products were amplified with 40 cycles of denaturation (95 °C, 15 s) and annealing and extension (60 °C, 60 s). The primers used for PCR were as follows: *hypoxanthine guanine phosphoribosyl transferase (Hprt)*, 5′-GCT CGA GAT GTC ATG AAG GAG-3′ and 5′-CAG CAG GTC AGC AAA GAA CTT-3′; *cathepsin K*, 5′-GGC TGT GGA GGC GGC TAT-3′ and 5′-AGA GTC AAT GCC TCC GTT CTG-3′; *matrix metalloproteinase 9 (mmp-9)*, 5′-AAA GAC CTG AAA ACC TCC AAC CT-3′ and 5′-GCC CGG GTG TAA CCA TAG C-3′ [25]; and *Gal-1*, 5′-CAG CAA CAA CCT GTG CCT AC-3′ and 5′-ACA ATG GTG TTG GCG TCT C-3′ [26]. Hprt was used as an internal control, and data were analyzed using the 2^−ΔΔCt^ method.

### 2.9. Resorption Assay

Differentiated cells on Osteo Assay Surface Stripwell Microplate (Corning) were removed by incubating the plate with 10% bleach. The plate was then washed with distilled water and air-dried. The dried plate was observed using an Olympus SZ61 stereo microscope (Olympus, Tokyo, Japan) equipped with a Swiftcam SC1603 digital camera (Swift, San Antonio, TX, USA), and images were captured using Swift imaging software (version 3.0). The area of the resorbed surface was measured using ImageJ software (ver. 1.54d) [24].

### 2.10. Oxidation of Recombinant Gal-1 Protein

Recombinant mGal-1 wild-type protein dissolved in PBS was treated with 10 mM H_2_O_2_ (Fujifilm Wako) for 2 h at 37 °C in the dark and then subjected to ultrafiltration using Amicon^®^ Ultra-15 filter devices (Merck Millipore, Burlington, MA, USA) to remove H_2_O_2_. The sugar-binding ability and oxidation state of the Cys residues in the prepared oxidized mGal-1 protein (Gal-1/Ox) were confirmed, and the oxidized mGal-1 was sterilized using filtration.

### 2.11. Hemagglutination Assay

The hemagglutination assay was performed as described previously [27]. Briefly, rabbit erythrocytes treated with trypsin and glutaraldehyde were mixed with mGal-1WT or mGal-1/Ox in 100 µL of PBS containing 50 µg/mL bovine serum albumin in a 96-well V-shaped titer plate. After incubating at room temperature for 1 h, images were captured using a digital camera.

### 2.12. Redox State Monitoring of Cys Residues

SulfoBiotics-Protein Redox State Monitoring Kit (Dojindo) was used to label the reduced (free) Cys residues of mGal-1 WT and mGal-1/Ox proteins. The labeling of the reduced Cys residue with Protein-SHifter was performed according to the manufacturer’s instructions. The labeled proteins were subjected to sodium dodecyl sulfate-polyacrylamide gel electrophoresis (SDS-PAGE) and coomassie brilliant blue staining using Bio-Safe™ Coomassie (Bio-Rad, Hercules, CA, USA). The redox state of Cys residues was monitored based on molecular weight because the labeling of one Protein-SHifter molecule resulted in a molecular shift of approximately 15 kDa.

### 2.13. Statistical Analysis

Microsoft Excel was used for the statistical analysis between two groups. Data are expressed as the mean ± standard deviation and were analyzed using Student’s *t*-test. For statistical analysis involving three or more groups, StatMate 3 software (ATOMS Inc., Tokyo, Japan) was employed. Significance of the variance was determined using one-way ANOVA, followed by Tukey’s multiple comparison test. *p*-values < 0.05 were considered statistically significant.

## 3. Results

### 3.1. Recombinant Gal-1 Protein Suppresses Osteoclastic Differentiation of Human PBMCs

To understand the effect of extracellular Gal-1 on osteoclast differentiation, we first investigated whether the addition of the recombinant Gal-1 protein suppressed osteoclastic differentiation of normal human PBMCs. PBMCs were treated with M-CSF and RANKL in the presence of the recombinant Gal-1 protein, allowed to differentiate, and subjected to a cell viability assay, TRAP staining, actin staining, and a resorption assay (Figure 1). The addition of the recombinant human Gal-1 protein did not affect cell viability but tended to reduce the number of TRAP-positive osteoclasts (Figure 1a,b). In addition, recombinant Gal-1 decreased the area surrounded by the actin ring and the area of the resorbed surface (Figure 1c,d). These results indicate that extracellular Gal-1 suppresses the osteoclastic differentiation of human PBMCs.

### 3.2. Recombinant Gal-1 Protein Suppresses the Osteoclastic Differentiation of Murine RAW264 Cells

To gain mechanistic insight into the effect of extracellular Gal-1 on osteoclastogenesis, we investigated whether recombinant Gal-1 also suppressed the differentiation of murine RAW264 cells, a well-known model cell line for osteoclast differentiation [18]. RAW264 cells were treated with RANKL in the presence of recombinant mGal-1 or heat-inactivated Gal-1, allowed to differentiate, and subjected to the real-time PCR analysis of osteoclast marker genes, TRAP staining, actin staining, and a resorption assay (Figure 2). The addition of the recombinant mGal-1 protein did not affect the RANKL-dependent expression of the osteoclast marker genes *cathepsin K* (Figure 2a) and *mmp-9* (Figure 2b). Gal-1 treatment reduced the number of TRAP-positive multinuclear cells, whereas heat-inactivated Gal-1 did not (Figure 2c). In addition, recombinant Gal-1 decreased the area surrounded by the actin ring and the area of the resorbed surface (Figure 2d,e). These results suggest that extracellular Gal-1 suppresses the osteoclastic differentiation of murine RAW264 cells.

### 3.3. Knockdown of Gal-1 Increases Formed TRAP-Positive Multinuclear Cells

RNAi experiments were performed to elucidate the effect of endogenous Gal-1 on osteoclast differentiation. It has been reported that Gal-1 mRNA is expressed in cells of monocyte/macrophage lineage, and its expression decreases upon differentiation into osteoclasts [16]. RAW264 cells were transfected with two different sequences of Gal-1 siRNAs (Gal-1 #1 and #2) or a negative control siRNA, and the effects of these siRNAs on gal-1 mRNA expression, cell viability, and the RANKL-dependent formation of TRAP-positive multinuclear cells were assessed (Figure 3). Transfection with each Gal-1 siRNA suppressed Gal-1 mRNA expression (Figure 3a). Each Gal-1 siRNA did not affect cell viability but increased the number of formed osteoclast-like cells (Figure 3b,c). These results suggest that endogenous Gal-1 may suppress the osteoclastic differentiation of RAW264 cells in an autocrine-like manner.

### 3.4. Reduced Form, Not Oxidized Form, of Gal-1 Suppresses the Formation of TRAP-Positive Multinuclear Cells

Extracellular Gal-1 has been reported to undergo oxidative inactivation, which results in the loss of its sugar-binding ability via the oxidation of its Cys residues and the formation of intra- and intermolecular disulfide bonds [10]. It has been reported that the oxidized form of Gal-1 functions in macrophage activation in a sugar-independent manner [13].

Therefore, we assessed the effect of the oxidized form of Gal-1 (Gal-1/Ox) on the osteoclastic differentiation of RAW264 cells. We prepared recombinant wild-type mGal-1 protein and oxidized it by treating it with H_2_O_2_. The properties of the prepared Gal-1/Ox were evaluated by performing a hemagglutination assay and redox-state monitoring (Figure 4a,b). The hemagglutination assay revealed that Gal-1/Ox exhibited little hemagglutinating activity, suggesting the loss of its sugar-binding ability. The redox state monitoring assay using Protein-SHifter and SDS-PAGE showed that the molecular weight of Gal-1/Ox was unaffected by the Protein SHifter treatment, whereas the molecular weight of wild-type Gal-1 shifted to approximately 110 kDa, which closely aligned with the theoretical value when all the cysteine residues of Gal-1 were labeled with Protein-SHifter. The molecular weight of Gal-1 was slightly less than 15 kDa and contained six cysteine residues, and the labeling of Protein-SHifter to one reduced cysteine residue resulted in a shift of approximately 15 kDa in the molecular weight. These results suggest the successful preparation of Gal-1/Ox. We assessed the effect of Gal-1/Ox on osteoclastic differentiation. RAW264 cells were treated with RANKL in the presence of recombinant mGal-1 or Gal-1/Ox, allowed to differentiate, and subjected to TRAP staining (Figure 4c). Gal-1 treatment suppressed the RANKL-dependent formation of TRAP-positive multinuclear cells, whereas Gal-1/Ox treatment did not. This result suggests that only the reduced form of the Gal-1 protein, which is not oxidized, affects osteoclast differentiation.

### 3.5. Gal-1 Suppresses the Formation of TRAP-Positive Multinuclear Cells in a β-Galactoside-dependent Manner

Gal-1 exhibits sugar (b–galactoside)-dependent and -independent functions, including binding pre-B-cell receptors and affecting their activation state [28,29]. We investigated whether the effect of Gal-1 on osteoclast differentiation was β-galactoside-dependent. RAW264 cells were treated with RANKL in the presence of recombinant mGal-1 and lactose, which contain a β-galactoside structure, or maltose, an osmotic control. After differentiation, the cells were subjected to TRAP staining (Figure 5). In the presence of lactose but not maltose, Gal-1 exerted little effect on TRAP-positive multinuclear cell formation. When lactose, which contains a β-galactoside structure, was added, it possibly inhibited the binding of Gal-1 to its ligand glycoconjugates on the surface of RAW264 cells. Consequently, the differentiation-inhibitory effect of Gal-1 was not observed. In contrast, maltose used as an osmotic control and lacking a β-galactoside structure did not inhibit the interaction between Gal-1 and its ligand glycoconjugates on the surface of RAW264 cells. Therefore, the differentiation-inhibitory effect of Gal-1 was observed. These results suggest that Gal-1 affects the osteoclastic differentiation of RAW264 cells in a β-galactoside-dependent manner.

## 4. Discussion

In the present study, we investigated the effects of extracellular Gal-1 on osteoclast differentiation and found that it suppressed the differentiation of human PBMCs and murine RAW264 cells. Also, we found that the reduced but not oxidized form of Gal-1 affected cell differentiation in a β-galactoside-dependent manner (Figure 4 and Figure 5). As Gal-1 exerted little effect on the expression of osteoclast marker genes, it could be considered that Gal-1 mainly affected the late phases of differentiation, including osteoclast cell fusion, actin ring formation, and bone resorption (Figure 1 and Figure 2). Although the detailed molecular mechanism underlying the effect of Gal-1 on differentiation remains unknown, to the best of our knowledge, this is the first study to report that extracellular Gal-1 suppresses osteoclastic bone resorption.

We found that extracellular Gal-1 tended to reduce the formed osteoclast number of human PBMCs but significantly suppressed actin ring formation and bone resorption (Figure 1). Similar results were obtained for RAW264 cells (Figure 2). However, Gal-1 did not affect the expression of osteoclast marker genes in RAW264 cells. We also investigated the effect of Gal-1 on the early phase of osteoclast differentiation signaling (Appendix A). Gal-1 treatment showed little effect on RANKL-dependent ERK1/2 phosphorylation (Appendix A). Furthermore, Gal-1 exhibited no significant effect on RANKL-dependent NFATc1 activation, specifically in terms of its nuclear localization (Appendix A). In the absence of RANKL stimulation, NFATc1 is primarily localized outside the nucleus. However, upon RANKL stimulation, NFATc1 was observed throughout the entire cell, including the nucleus, regardless of the presence of Gal-1. Given these results, it could be considered that Gal-1 mainly affects the late phases of osteoclast differentiation, such as cell fusion, actin ring formation, and resorption but not the early phases, such as the expression of osteoclast marker genes. Gal-1 knockdown in RAW264 cells increased the number of TRAP-positive multinuclear cells (Figure 3). This result is somewhat inconsistent with the results observed in Gal-1 knockout mice; Muller et al. reported that primary osteoclast cultures of Gal-1 knockout mice showed higher bone resorption activity but little difference in the number of TRAP-positive multinuclear cells compared with those shown by the wild type [16]. There may be a slight difference in the effect of Gal-1 on osteoclasts due to the cells used and differentiation conditions. However, the results presented by Muller et al. [16] and the present results are consistent with the suppressive effect of Gal-1 on the most important function of osteoclasts, namely, bone resorption. These findings support the hypothesis that Gal-1 primarily affects the late phase of osteoclast differentiation.

Extracellular Gal-1 suppressed the formation of multinuclear osteoclasts, actin rings, and bone resorption (Figure 1 and Figure 2). Gal-1 plays a role in trophoblast cell fusion via its interaction with syncytin-2 [30,31]. Syncytin-1, not syncytin-2, is involved in osteoclast fusion [32]. To the best of our knowledge, the role of syncytin-2 in osteoclast fusion has not been reported. Gal-1 may affect osteoclast cell fusion via its interaction with syncytin family proteins or other proteins involved in cell fusion, such as osteoclast stimulatory transmembrane protein (OC-STAMP), dendritic cell-specific transmembrane protein (DC-STAMP), ATP6v0d2, and integrins [33]. Our preliminary experiments revealed that one of the Gal-1 binding receptors of RAW264 cells is integrin β2 (unpublished data), which has already been reported as a binding partner for Gal-1 [34]. As integrin β2 is involved in osteoclast differentiation [35], it could be hypothesized that Gal-1 affected differentiation through its binding to integrin β2. On the other hand, if Gal-1 mainly affects the late phase of osteoclast differentiation, it should be noted that Gal-1 binds integrin β3 [36,37]. Integrin β3 is involved in the binding of osteoclasts on the bone surface and affects the late phase of differentiation; its knockout mouse model showed a reduced number of osteoclasts, abnormal actin ring formation, and lower bone resorption activity than the wild type [38]. It could also be possible that Gal-1 bound to integrin β3 and affected osteoclast differentiation. Such a line of study is important to clarify the detailed mechanism underlying the effect of extracellular Gal-1 on osteoclasts, and we are currently planning to pursue it.

Xu et al. reported that decreased bone mass was correlated with decreased serum Gal-1 levels [17]. The origin of Gal-1 in the serum could be various tissues, including muscles. Gal-1 has been reported to be involved in muscle cell differentiation and highly expressed in muscles [37,39,40,41]. In recent years, increasing attention has been paid to the role of muscles and bones as endocrine organs, and molecules secreted by each organ affect each other, particularly in the context of conditions such as sarcopenia [42,43]. Human muscle progenitor cells secrete Gal-1, which exerts an immunosuppressive effect [44]. Gal-1 might be secreted from muscle cells as a myokine, a skeletal muscle-derived humoral cytokine, and affect bone metabolism. Taken together, the reduction in muscle mass due to aging probably leads to a decrease in circulating Gal-1 levels. This, in turn, may inhibit bone formation by affecting osteoclasts and osteoblasts.

The findings of this study suggest that extracellular Gal-1 in its reduced form inhibits osteoclastic bone resorption. This insight, in conjunction with a report showing that Gal-1 promotes bone formation [15], may be beneficial for addressing conditions such as osteoporosis and sarcopenia, as well as contributing to a better understanding of musculoskeletal interactions, particularly those relevant in an aging society.

## Figures and Tables

**Figure 1 biomolecules-14-00121-f001:**
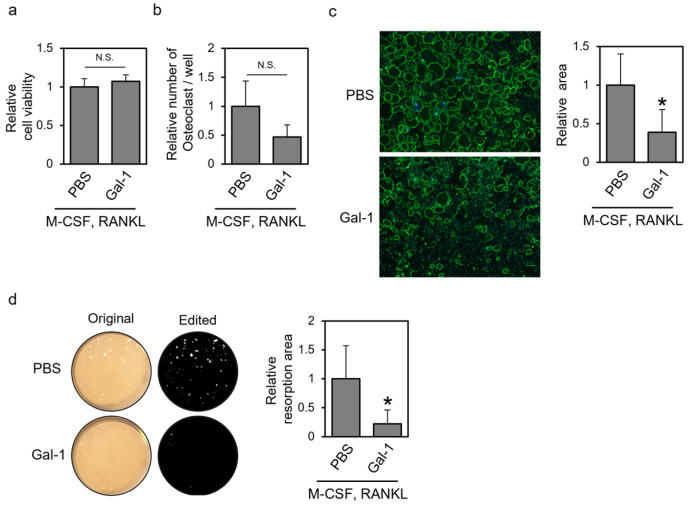
Recombinant galectin-1 (Gal-1) protein suppresses the osteoclastic differentiation of human peripheral blood mononuclear cells (PBMCs). Human normal PBMCs were treated with macrophage colony-stimulating factor (M-CSF) and receptor activator of nuclear factor-κB ligand (RANKL) in the presence of the recombinant hGal-1C2S protein for 7 days for cell viability assay and tartrate-resistant acid phosphatase (TRAP) staining or 14 days for actin staining and resorption assay. The cells were then subjected to (**a**) cell viability assay, (**b**) TRAP staining, (**c**) actin staining, and (**d**) resorption assay. Representative images of actin staining (actin, green; nuclei, blue) and resorption assay are shown. In the edited images of the resorption assay, resorption areas are highlighted in white. Data are expressed as the mean ± standard deviation (S.D.) (*n* = 4 or 8). * *p* < 0.05 vs. control (PBS with M-CSF and RANKL). N.S. means not significant.

**Figure 2 biomolecules-14-00121-f002:**
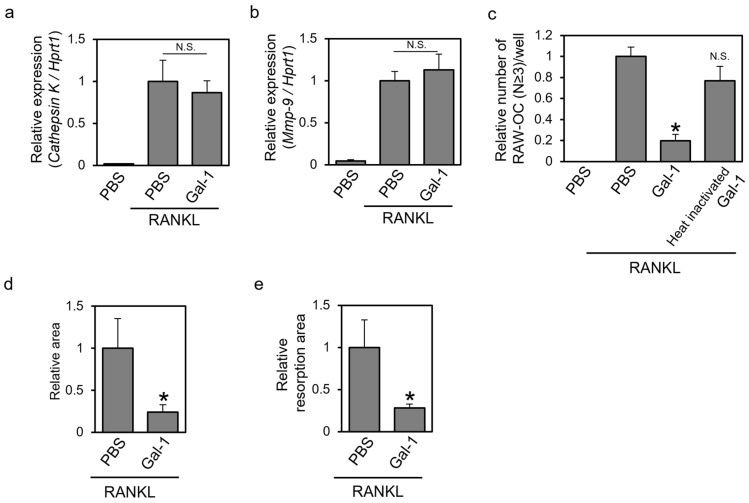
Recombinant Gal-1 protein suppresses the osteoclastic differentiation of murine RAW264 cells. Murine RAW264 cells were treated with RANKL in the presence of recombinant mGal-1C2S protein or heat-inactivated mGal-1C2S protein for 4 or 14 days. The cells were then subjected to (**a**,**b**) real-time polymerase chain reaction (PCR), (**c**) TRAP staining, (**d**) actin staining, and (**e**) resorption assay. Only the resorption assay was performed after a 14-day culture period. RAW-OC means RAW264-derived osteoclast-like TRAP-positive multinuclear cell. Data are expressed as the mean ± S.D (*n* = 3, 4 or 8). * *p* < 0.05 vs. control (PBS with RANKL). N.S. means not significant.

**Figure 3 biomolecules-14-00121-f003:**
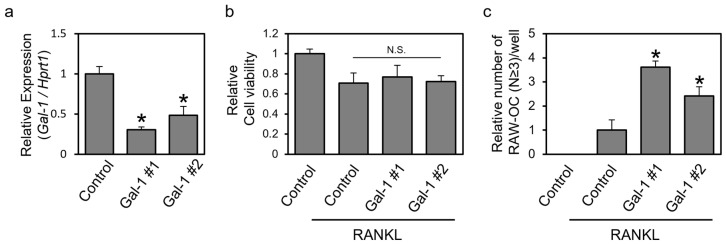
The knockdown of Gal-1 increases formed TRAP-positive multinuclear cells. (**a**) RAW264 cells were transfected with indicated small interfering RNAs (siRNAs). After incubation for 24 h, the cells were collected and examined for Gal1 mRNA expression using real-time PCR. (**b**,**c**) RAW264 cells were transfected with indicated siRNAs and treated with RANKL. After incubation for 4 days, the cells were subjected to (**b**) cell viability assay and (**c**) TRAP staining. Data are expressed as the mean ± S.D. *(n* = 3 or 4). * *p* < 0.05 vs. control (negative control siRNA). N.S. means not significant.

**Figure 4 biomolecules-14-00121-f004:**
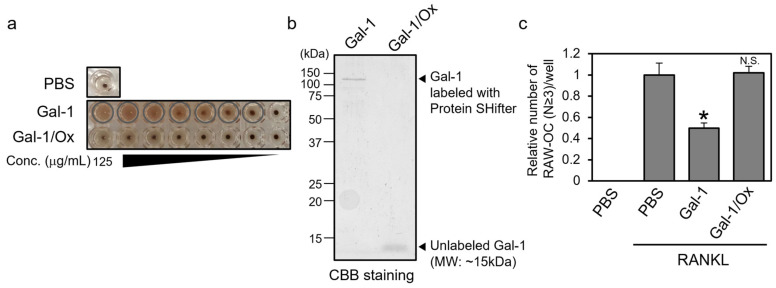
The reduced form, not the oxidized form, of Gal-1 suppresses the formation of TRAP-positive multinuclear cells. (**a**) Indicated final concentrations (set at a maximum of 125 µg/mL, a 2-fold dilution series was prepared) of Gal proteins were mixed with rabbit erythrocytes and incubated for one hour. Images were captured using a digital camera. The wells corresponding to agglutination are indicated by grey circles. (**b**) Wild-type Gal-1 or Gal-1/Ox proteins were treated with Protein SHifter and subjected to sodium dodecyl sulfate–polyacrylamide gel electrophoresis. (**c**) RAW264 cells were treated with RANKL in the presence of recombinant Gal-1 or Gal-1/Ox for 4 days. The cells were then subjected to TRAP staining. Data are expressed as the mean ± S.D (*n* = 4). * *p* < 0.05 vs. control (PBS with RANKL). N.S. means not significant.

**Figure 5 biomolecules-14-00121-f005:**
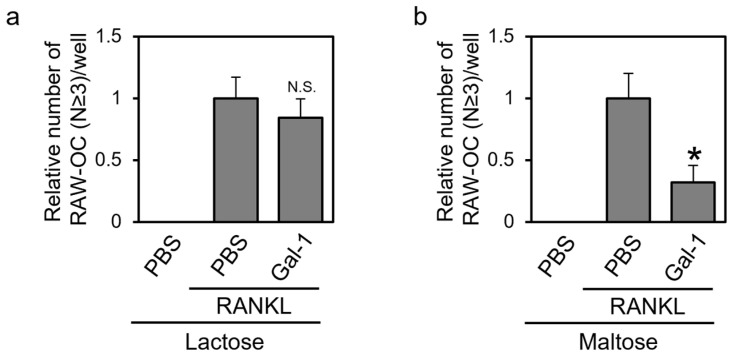
Gal-1 suppresses the formation of TRAP-positive multinuclear cells in a β-galactoside-dependent manner. Murine RAW264 cells were treated with RANKL in the presence of recombinant mGal-1C2S protein and competitive sugar, lactose (**a**) or maltose (**b**) for 4 days. The cells were then subjected to TRAP staining. Data are expressed as the mean ± S.D. (*n* = 4). * *p* < 0.05 vs. control (PBS with RANKL). N.S. means not significant.

## Data Availability

The data generated or analyzed during this study are included in this article.

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
