# Peer review of "Reduced form of Galectin-1 Suppresses Osteoclastic Differentiation of Human Peripheral Blood Mononuclear Cells and Murine RAW264 Cells In Vitro"

_biomolecules, 2024, doi:10.3390/biom14010121_

Round 1

Reviewer 1 Report

Comments and Suggestions for Authors

In this article, the author reveals that extracellular Gal-1 inhibits osteoclast formation and activity in human PBMCs and mouse RAW264 cells, impacting differentiation through a β-galactoside-dependent mechanism. Gal-1 knockdown enhances osteoclast-like cell formation, while the presence of oxidized Gal-1 and lactose negates its inhibitory effects. Several major points need addressing:

  1. If the graph shows a comparison of three or more groups, the author must conduct the analysis using ANOVA and describe it in the Materials and Methods section.
  2. In Fig. 1, add the statistical mark 'Not significant' as well.
  3. In Fig. 3, it is noteworthy that the authors did not investigate how Gal-1 expression is regulated during osteoclast differentiation.
  4. A literature search has revealed that Gal-1 affects IκB, ERK, and AKT, which are important signals for osteoclastogenesis. Therefore, the authors should investigate whether Gal-1 affects these signals. Additionally, they should examine the impact of Gal-1 on osteoclast differentiation marker genes, both with Galectin-1 and Gal-1/ox.

In Fig. 4, I recommend changing the title to 'The Reduced form, not the oxidized form, of Gal-1 suppresses the formation of TRAP-Positive multinuclear cells' for clarity.

Author Response

RESPONSES TO COMMENTS BY REVIEWER #1

We thank you for your favorable and thoughtful comments on our manuscript, which have enriched the manuscript and produced a better and more balanced account of the research. Our responses are described below. We hope that the revised manuscript is now suitable for publication in Biomolecules.

Comment #1:

If the graph shows a comparison of three or more groups, the author must conduct the analysis using ANOVA and describe it in the Materials and Methods section.

Response:

According to your suggestion, when dealing with the data containing three or more groups, we conducted Tukey's test instead of a t-test. This information has been added to the Materials and Methods section (Lines 204-208).

Comment #2:

In Fig. 1, add the statistical mark 'Not significant' as well.

Response:

We have added the statistical mark for 'Not significant' in Figure 1. We have also added the mark in the other figures, and changed the figure legends accordingly.

Comment #3:

In Fig. 3, it is noteworthy that the authors did not investigate how Gal-1 expression is regulated during osteoclast differentiation.

Response:

We have added the description about Gal-1 mRNA expression during differentiation in Lines 257-259.

Comment #4:

A literature search has revealed that Gal-1 affects IκB, ERK, and AKT, which are important signals for osteoclastogenesis. Therefore, the authors should investigate whether Gal-1 affects these signals. Additionally, they should examine the impact of Gal-1 on osteoclast differentiation marker genes, both with Galectin-1 and Gal-1/ox.

Response:

We appreciate your thoughtful comments. We had already investigated the effect of Gal-1 on RANKL-dependent activation of ERK1/2 and NFATc1 and found Gal-1 has little effect on them, although these findings were undescribed in the previously submitted version of our manuscript. Therefore, in response to your suggestion, we have added these data as a supplemental figure (Figure S1). Descriptions related to the data in the figure have been added to the Discussion section (Lines 348-355), along with corresponding explanations (Lines 404-406).

The effect of Gal-1 on the expression of differentiation marker genes has already been described in Figure 2a and b, showing Gal-1 has little effect on the expression of the differentiation marker genes. These results suggest that Gal-1 has little effect on the early phase of osteoclast differentiation signaling, such as the activation of NF-kB, ERK, AKT, and NFATc1. On the other hand, regarding Gal-1/Ox, it did not exhibit any significant impact on multinucleated osteoclast formation, actin ring formation, and bone resorption. Given these observations, it strongly suggests that Gal-1/Ox does not influence the early signals of osteoclast differentiation. Consequently, we did not perform similar experiments.

Comment #5:

In Fig. 4, I recommend changing the title to 'The Reduced form, not the oxidized form, of Gal-1 suppresses the formation of TRAP-Positive multinuclear cells' for clarity.

Response:

We have changed the title of the Figure 4 as per your suggestion.

Reviewer 2 Report

Comments and Suggestions for Authors

Takeuchi et al. present evidence that recombinant galectin-1 inhibits osteoclast differentiation and bone resorption, which could be an interesting candidate for understanding the Gal-1 role in the osteoporosis model.

The novelty of the current study is that it demonstrates that extracellular Gal-1 suppresses (reduced, not oxidized of, Gal-1) the differentiation of osteoclast in a b-galactoside-dependent manner. This manuscript is well-written and describes each point very well. I have minor comments which are essential to address. 

1-The author used a high concentration of sRANKL (250ng/mL) in mouse macrophage RAW264. The author should mention research article references that have been used 250ng/mL for RAW cell differentiation.

2-Fig. 3a says that relative expression. The author should mention the mRNA expression of Gal-1 in the figure and figure legend, which would be more apparent.

3-Line 308-311- The author says that In the presence of lactose, but not maltose, Gal-1 exerted little effect on TRAP-positive multinuclear cell formation. These results suggest that Gal-1 affects the osteoclastic differentiation of RAW264 cells in a b-galactoside-dependent manner. In Fig 5, in the presence of maltose, there is a significant decrease in TRAP-positive multinuclear cell formation in the presence of gal-1. The author should explain/discuss more about maltose's role in osteoclastogenesis and how it affects osteoclastogenesis.

Author Response

RESPONSES TO COMMENTS BY REVIEWER #2

We thank you for your favorable and thoughtful comments on our manuscript, which have enriched the manuscript and produced a better and more balanced account of the research. Our responses are described below. We hope that the revised manuscript is now suitable for publication in Biomolecules.

Comment #1:

The author used a high concentration of sRANKL (250ng/mL) in mouse macrophage RAW264. The author should mention research article references that have been used 250ng/mL for RAW cell differentiation.

Response:

First, we would like to apologize for the insufficient description regarding the RANKL used in our study. We used GST-tagged soluble RANKL obtained from Oriental Yeast. Unlike sRANKL from PeproTech or R&D Systems, this particular sRANKL carries a GST tag. To ensure clarity on the use of GST-tagged sRANKL, we have amended the relevant description in the Materials and Methods (Line 104, 113).

              Regarding the concentration of GST-tagged sRANKL used in this study for the differentiation of RAW264 cells, it is higher than the concentrations typically used for osteoclast differentiation. This is due to the difference in molecular weight between GST-tagged sRANKL, which we used, and untagged sRANKL (Oriental Yeast's GST-tagged sRANKL has a molecular weight of 47 kDa, whereas, for example, PeproTech's sRANKL is 20 kDa). We have consistently used 250 ng/mL GST-tagged sRANKL for the differentiation of RAW264 cells. We have cited a new reference, REF23 (Line 113; Lines 470-472), regarding this matter, and have changed the subsequent reference numbers accordingly. The concentration of GST-tagged sRANKL used for differentiation was determined through our preliminary experiment to optimize the differentiation condition, where we found 250 ng/mL of GST-tagged sRANKL provide the most efficient differentiation, minimal cytotoxicity, and consistent results. We have documented this in the Materials and Methods of REF23, stating, “RANKL (GST-tagged sRANKL) concentration used in this study was optimized by our preliminary experiments (data not shown).”

Comment #2:

Fig. 3a says that relative expression. The author should mention the mRNA expression of Gal-1 in the figure and figure legend, which would be more apparent.

Response:

In response to your comment and considering Figure 2, we have updated the Y-axis label for Figure 3a to 'Relative Expression (Gal-1 / Hprt1).' We have also added relevant description in the figure legend (Line 272).

Comment #3:

Line 308-311- The author says that In the presence of lactose, but not maltose, Gal-1 exerted little effect on TRAP-positive multinuclear cell formation. These results suggest that Gal-1 affects the osteoclastic differentiation of RAW264 cells in a b-galactoside-dependent manner. In Fig 5, in the presence of maltose, there is a significant decrease in TRAP-positive multinuclear cell formation in the presence of gal-1. The author should explain/discuss more about maltose's role in osteoclastogenesis and how it affects osteoclastogenesis.

Response:

We apologize for the insufficient explanation regarding the experiments in Figure 5. We have described additional explanations and interpretations, including the effect of maltose, on the experimental results (Lines 319-324).

Reviewer 3 Report

Comments and Suggestions for Authors

The manuscript “Extracellular galectin-1 suppresses osteoclastic differentiation …” deals with the effect of extracellular galectin Gal-1 in reduced or oxidized from on osteoclastic differentiation plus bone resorption activity of PBMCs. The authors studied the effect of Gal-1 on two different types of cells, whereby the effect observed with RAW264 cells, an often-used model cell line for osteoclast differentiation was particularly convincing.

The authors used Protein Shifter, a not so commonly known assay to characterize native (reduced) and S-S oxidized Gal-1.

The experiments appear to be thoroughly planned and conducted. The results are presented in a very clear manner. Simple but – as far as I can rate – appropriate statistics.

The research question is clear and worth being dealt with. The sequence of chapters and experiments is convincing.

The long, but definitely not unduly lengthy discussion attests the deep familiarity of the authors with their subject by discussing results of other groups and the manifold contact points of Gal-1 with ligands etc. 

Just a few minor comments:

l 272: I would not write “oxidative form of Gal-1”   if oxidized form is meant.

l 289:  that ONLY ? the reduced form

Author Response

RESPONSES TO COMMENTS BY REVIEWER #3

We thank you for your favorable and thoughtful comments on our manuscript, which have enriched the manuscript and produced a better and more balanced account of the research. Our responses are described below. We hope that the revised manuscript is now suitable for publication in Biomolecules.

Comment #1:

l 272: I would not write “oxidative form of Gal-1”   if oxidized form is meant.

Response:

We have changed the description as per your suggestion (Line 279, 281).

Comment #2:

l 289:  that ONLY ? the reduced form

Response:

We have changed the description as per your suggestion (Line 298).

Reviewer 4 Report

Comments and Suggestions for Authors

Galectin-1 is a lectin which function on osteoclast differentiation, including bone resorption activity remains unclear. Authors of this manuscript reported the effect of galectin-1 protein on the osteoclastic differentiation. Their results showed that treatment of recombinant galectin-1 protein suppressed the osteoclastic differentiation of human peripheral blood mononuclear cells and murine RAW264 cells in a beta-Galgalactoside-dependent manner in vitro. And, galectin-1 proteins in its reduced form, not in oxidized form, affected osteoclast differentiation. These results were interesting, which may contribute to understanding the regulation of bone metabolism. But, although some hypothesis on the molecular mechanism during the process above have been mentioned in the manuscript, there is no detailed evidences presented. Therefore, more and deep investigation are surely needed further.

Some modification are suggested:

Line 2, “Extracellular” in this manuscript means “in vitro”.

Line 70, “serum Gal-1, i.e., extracellular Gal-1,” should be modified. “serum Gal-1” is different from “extracellular Gal-1”.

Fig.4(b), MW of Gal-1 (unlabled) is 11-12kDa? 15kDa?

Comments on the Quality of English Language

no comments

Author Response

RESPONSES TO COMMENTS BY REVIEWER #4

We thank you for your favorable and thoughtful comments on our manuscript, which have enriched the manuscript and produced a better and more balanced account of the research. Our responses are described below. We hope that the revised manuscript is now suitable for publication in Biomolecules.

Comment #1:

These results were interesting, which may contribute to understanding the regulation of bone metabolism. But, although some hypothesis on the molecular mechanism during the process above have been mentioned in the manuscript, there is no detailed evidences presented. Therefore, more and deep investigation are surely needed further.

Response:

Thank you for your positive feedback. We also believe that more in-depth investigations are necessary in the future. We are currently planning to test the hypothesis outlined in the Discussion. To emphasize this point, we have added a statement to the Discussion section indicating our ongoing experiments in this direction (Lines 386-387).

Comment #2:

Line 2, “Extracellular” in this manuscript means “in vitro”.

Response:

In response to the editor’s suggestion, we have revised the manuscript title and have deleted “Extracellular.” As per your suggestion, we have added “in vitro” at the end of the title (Line 4).

Comment #3:

Line 70, “serum Gal-1, i.e., extracellular Gal-1,” should be modified. “serum Gal-1” is different from “extracellular Gal-1”.

Response:

As per your suggestion, we have removed “serum Gal-1” and now simply refer to it as “extracellular Gal-1.” (Lines 70-71)

Comment #4:

Fig.4(b), MW of Gal-1 (unlabled) is 11-12kDa? 15kDa?

Response:

The theoretical molecular weight of Gal-1 is approximately 15 kDa. However, on SDS-PAGE, a band is commonly observed around 14 kDa. Considering your feedback, we have modified the description of the arrow indicating unmodified Gal-1 in Figure 4b to read 'Unlabeled Gal-1 (MW: ~15 kDa)' to reflect the theoretical molecular weight of Gal-1.

Round 2

Reviewer 1 Report

Comments and Suggestions for Authors

None